# Net Zero Buildings—A Framework for an Integrated Policy in Chile

**María Beatriz Piderit [1],\*, Franklin Vivanco [1], Geoffrey van Moeseke [2] and Shady Attia [3]**

[1] Department of Design and Theory of Architecture, University of Bío-Bío, Av. Collao 1202, Concepción, Chile; frankvivanco@gmail.com

[2] Architecture et Climat—Faculté d'Architecture, d'Ingénierie Architecturale, d'Urbanisme (LOCI) Université Catholique de Louvain, 1348 Ottignies-Louvain-la-Neuve, Belgium; geoffrey.vanmoeseke@uclouvain.be

[3] Sustainable Building Design Lab, Department UEE, Applied Sciences, Université de Liège, 4000 Liège, Belgium; shady.attia@uliege.be

\* Correspondance: mpiderit@ubiobio.cl

**Abstract:** The potential of carbon dioxide emissions mitigation in the building sector can be achieved through energy policies, progressive goals, and support systems to attain sustainable constructions that guarantee the reduction of emissions. Net-Zero Energy Buildings (NZEB) is a concept that allows moving forward to neutralize buildings' carbon emissions. This has been demonstrated by more industrial countries which have set goals and challenges to progressively approach an energy neutrality balance for buildings. Therefore, the target of this research is to define a framework for a new standard to reach NZEB in Chile. Firstly, an exhaustive review of the energy policies, NZEB definitions, and components of an NZEB system took place. Secondly, focus group discussions with local and international professionals from the building sector were organized to define a vision, opportunities, and potential measures with a focus on policies, to implement and develop local technologies for NZEB buildings in Chile. The study identifies the need to advance public policies to achieve an integrated policy for the implementation of energy neutral concept buildings. Finally, the paper presents a NZEB standard framework, including key performance indicators and suggested performance metrics thresholds.

**Keywords:** energy efficiency; energy policy; nearly zero energy building; energy demand; thermal comfort; Latin America

## 1. Introduction

In recent years, Europe has begun implementing the concept of NZEB in policies for new and existing buildings in the private and public sectors, and for residential and non-residential buildings [1,2]. The European Economic Community is a pioneer in setting goals regarding the implementation of Net-Zero Energy Buildings (NZEB), for all EU member states, having the first relevant deadline set for 2020. For example, the primary energy target values vary from the most ambitious at 20 kWh/m$^2$/year to 180 kWh/m$^2$/year in residential buildings. As shown in Table 1, France, among the more developed countries in Europe, has set an energy performance target in its building codes of 70 kWh/m$^2$/year for heating, domestic hot water (DHW), lighting, cooling, and auxiliary systems for non-residential buildings. This standard varies by climatic zone and altitude from 70 kWh/m$^2$/year to 110 kWh/m$^2$/year. They have proposed as a goal for 2020 that all new buildings must be energy positive; i.e., that they produce more energy than the energy they demand. Denmark, however, has established a national NZEB definition through a roadmap for 2020. This roadmap progressively sets the energy performance until reaching the strictest levels, starting from the

basic standard, passing through milestones to 2015 and 2020 [3]. On the other hand, in the region of Brussels in Belgium and since 2015 and onwards, all new public and residential buildings must have a primary energy demand performance close to a PassivHaus standard [4], considering a performance of 45 kWh/m$^2$/year for auxiliary energy and 15 kwh/m$^2$/year for heating and cooling. In general, it is felt that the benefits of setting these goals will only be seen between 2020 and 2050 [5].

Table 1. Performance target for nZEB definition in EU members [5,6].

| Country | Non-Residential Buildings (kWh/m$^2$/Year) | | Notes |
|---|---|---|---|
| | **New** | **Existing** | |
| Sweden | 30–105 | ND | Depending on the building type and climate |
| Spain | 45–60 | 120 | The proposed indicators aim to define maximum net PE use, maximum total PE use and minimum renewable contribution. |
| Romania | 50–102 | 120–400 | Depending on the building type and climate |
| France | 70–110 | ND | Depending on the climate |
| Denmark | 25 | 25 | Included: Heating, cooling, ventilation, DHW, lighting |
| Bulgaria | 40–60 | 40–60 | - |
| Belgium (Brussels) | 90 | 108 | Included: Heating, DHW and appliances |
| Austria | 170 | 250 | from (2021) |

Despite the efforts made, the implementation of the NZEB concept in Europe has been slow, and some countries have not yet been capable of establishing a definition of the concept within their regulation and in practice [7]. In Southern countries of Europe, there is a very slow advancement regarding the NZEB goal for 2020. In the case of Romania and Portugal, both countries have problems such as (1) a lack of professional knowledge about the design and construction of NZEBs, (2) a lack of local construction materials to reach a high standard, and (3) a lack of locally manufactured HVAC equipment which allows for high energy performance [7–9]. The user's behavior and the low purchase power mean that potential NZEB occupants prefer passive solutions rather than ones which demand high-tech equipment. Those types of findings and learned lessons presented above, contribute to a better understanding of NZEB challenges to better address them in the Chilean context.

Chile has focused, during the last 30 years, on closing the housing gap by creating policies and subsidy programs to reduce the market deficit [10]. However, the increasing expansion of the residential sector, the economic growth, joining the Organization for Economic Cooperation and Development (OECD), and signing the Paris Agreement, has created new legal obligations [11]. Chile must commit to higher energy efficiency and carbon dioxide emissions reduction targets for its construction sector. So far, the Chilean government succeeded to create energy efficacy regulations for public buildings, for example the TDRe (Chilean Reference Standard for Thermal Comfort and Energy Efficiency in Buildings) and the Sustainable Buildings certification system (CES) [12]. Both standard and certification requirements are mainly prescriptive and are not obligatory to implement. On the environmental side, an important step has been made through the Atmospheric Decontamination Plans (PDA). Because in Chile, wood burning stoves are commonly used and as a consequent, occupants suffer from high carbon monoxide, nitric oxide, nitrogen dioxide, and suspended particles, including benzopyrene [13]. However, the PDA is limited to some cities. This shows a shortage in the regulations and standards landscape that should set performance values of buildings during construction and operation in association with energy consumption carbon dioxide emissions [14–16]. The current standards and certificates do not address the NZEB challenge. There is a need to provide a legal framework for NZEB that defines their performance criteria and performance thresholds. Indeed, Chile has defined goals that are stated in its 2050 Energy Strategy [17] with concrete goals set for 2035 and 2050. However, the 2050 defined goal for the construction sector does articulate how those targets will be achieved.

To address the previously identified barriers, we aim to identify the gaps related to NZEB implementation in Chile, and to provide a framework to create a Chilean energy standard for NZEB. The study investigates the components and indicators required to construct a new framework. The

proposed framework is expected to be part of an integrated policy for Chile that articulates specific performance criteria and qualitative technical characteristics for NZEB. The added value of the paper relates to providing a broad overview on the challenges of NZEB implementation in Chile while bringing insights from EU member states, which has not been done before. Also, the paper identifies an initial framework for a NZEB standard that was validated based on a qualitative approach. Accordingly, the framework provides an overview for best practices in European countries regarding NZEB, to bridge the knowledge gap in Chile [7,8,18,19]. The overarching aim is to increase the NZEB market uptake in Chile and potentially in Latin America.

Major components of the paper include a literature review that covers more than 40 publications, followed by the results of focus groups discussions (FGD). The literature review allowed us to identify the definitions and functions of NZEB and to propose a framework for a new standard associated with identifying the key performance indicators (KPI) related to the energy balance. This was followed by the results of FGD that were conducted to gain a deeper understanding from international and local experts on the performance expectations of NZEB.

## 2. Methodology

In this section, we present the research methodology, including the study concept. Like the work of [20,21], our research methodology combines a literature review and focus group discussions (FGD). The concept of this study was built around three stages in the context of developing an integrated policy framework for NZEB in Chile. The study followed three stages for data collection and validation for the proposed framework. Figure 1 illustrates a detailed study conceptual framework of the research endeavor.

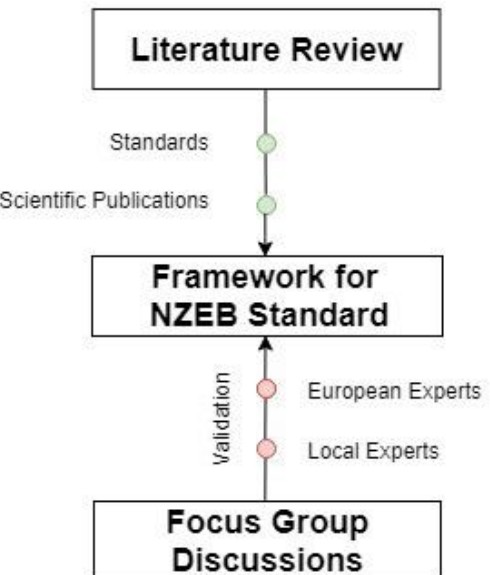

**Figure 1.** Study conceptual framework.

### 2.1. Literature Review

The first stage of our research methodology was based on a detailed literature review of an extensive number of publications. The literature review was prepared by exploring resources that refer to the concept and to building energy policies. The publications included standards, technical codes, books, manuals, conference materials, and scientific magazine articles. For the first level of research we checked topics regarding the definitions of the NZEB concept in *Web of Science*. More than 200 publications were found as a result. Then, the publications information was imported to the software HistCite for analysis. Results were grouped under two categories: definitions and design

principles (see Sections 3.1 and 3.3). This provided an overview of advances and evolutions of the NZEB concept mainly in the European market.

Moreover, a review of Chilean national and regional code and standards took place with a focus on residential buildings. The research was performed based on the information and documentation provided by the following private and public organization:

(1)    AChEE (Chilean Energy Efficiency Agency)
(2)    MOE (Ministry of Energy in Chile)
(3)    MOP EE (Eficiencia Energetica) Ministry of Public Works
(4)    Passivhaus Chile

### 2.2. Framework Development

The second stage of research involved the classification and screening of the literature review findings in order to translate them into a framework. The KPI's of NZEB were identified through the detailed analysis of the publications. Then, we began to articulate and define the major components and requirements of a potential Chilean standard. All information collected from the literature review was processed to develop the domains and items of the proposed standard framework. The authors crossed the policy landscape of energy efficiency in Chile against the European policy landscape for NZEB. This was followed by a qualitative evaluation of the potential and requirements of introducing NZEB in Chile.

### 2.3. Focus Group Discussions

The focus groups were administered as a collective exercise. With the guidance of an animator, the experts participated in identifying the barriers in Chile and elaborate with the European experts a model for a new NZEB framework for Chile. Two Focus Group Discussions (FGD) took place with more than 60 building professionals. The FGD are part of a project that aims to modernize the building regulations landscape in Chile [22]. The FGD allowed us to test and validate our framework, as well as to assess the potential market uptake of NZEB in Chile.

As a preparation step for the FGD, the literature review results were shared with the experts prior to the discussions. The literature review helped to identify the crucial themes and potential application requirements for NZEB in Chile. Therefore, the FGDs were organized around five round-table groups, guided by an animator to assess the framework regarding:

(1)    Technological Approach
(2)    Energy Efficiency and Primary Energy Requirements
(3)    Comfort Requirements
(4)    Renewables Share
(5)    Construction Quality

The design of the FGD required organizing two rounds of discussions. The first round of discussion would focus on comparing the market and policy landscape in Europe and Chile. This step is useful to identify dysfunctions in energy efficiency policies, and to understand the gaps between its theoretical implantation and reality. The second round of discussion would focus on developing and validating a novel standard framework for NZEB in Chile. The FGD results are presented in Section 3.3, allowing for the identification of future developments. The discussions helped to contextualize the framework for the new standard for NZEB in Chile. Also, the FGDs identified the challenges to make NZEB a mainstream in the construction sector and to align with the energy policies included in ENERGIA 2050 Strategy [17].

Local experts were recruited from different domains representing different stakeholders involved in the building construction industry in Chile. The inclusion criteria of experts to be admitted into FGD were made to cover the construction industry, the Chilean state, building professionals, and

building researchers. A convenience in sampling was used to recruit participants. Three European experts representing Germany, Spain, and Belgium took place during the two FGD rounds. The FGD were held in January and July 2017 in Spanish in the Bio-Bio University in Conception, Chile. A total of 60 experts participated in the development of discussions. In addition to the animator guiding the group activity and discussion, a moderator and two observers were present to assure that each participant had an equal voice and that all nuances were perceived. Also, a focus group guide based on the literature review described for experts the literature findings and suggested standard framework. The guide served as a reference to structure the FGD. Participants were also provided with a short, one-page summary of the literature review to help them understand the discussion topics.

Participants were grouped around five tables and asked to validate the literature findings and novel standard framework. The discussions were stopped when each team on every table agreed that saturation of information had been reached. Finally, the focus group outputs were analyzed. The common elements between the outputs of groups were transcribed into a report.

## 3. Results

### 3.1. NZEB Defintions

The definition of NZEB has evolved over time depending on different realities. This concept is greatly influenced by the socio-economic and political context of each country, which means there is no common definition. For this reason, adopting a definition implies a process of reflection and study about the parameters and factors that can be feasibly included in this. There are several studies that reviewed the definitions of NZEB [23–26]. The earliest definitions of NZEB is the one of Torcellini et al. [27] and later the work of D'Agosto et al. [5], who defined NZEB as "*has a very high energy performance with almost zero or very little energy required, covered mainly by renewable energy sources, including renewable energy sources produced onsite or nearby.*" One of the most influential contributions to define NZEB was the work of the International Energy Agency Annex 40 that grouped different researchers to define a framework for NZEB definitions [23]. As shown in Figure 2, researcher in Annex 40 framed the NZEB definition with different energy use boundaries. We can learn from their work that the core of the *net-zero energy* concept is the energy balance to offset the required energy. To achieve the energy balance, we firstly reduce the energy demand, and secondly, generate the energy needed to supply the demand [24]. The relationship with the grid is in this context is relevant because NZEB must be grid connected [24].

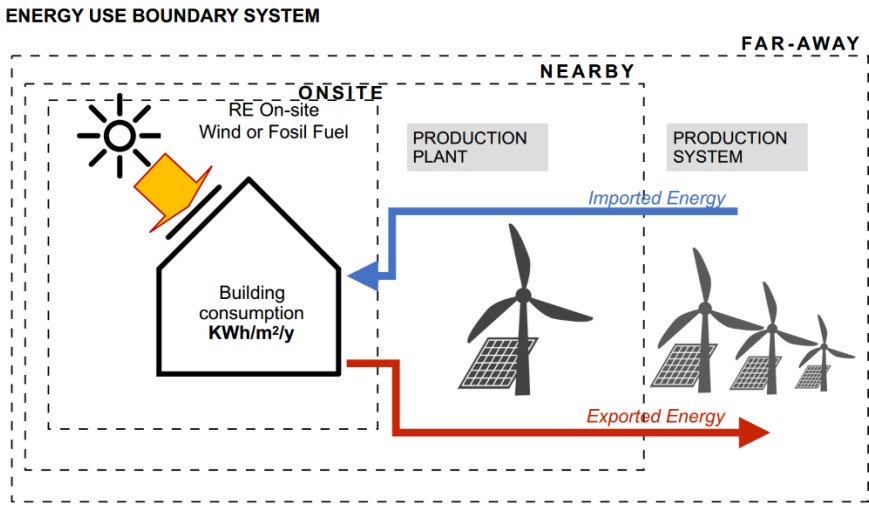

**Figure 2.** Energy generation diagram for the energy balance of a zero-energy building.

Surprisingly, with the evolution of the definition, from 2006 until 2013, a new definition emerged, namely the *nearly zero energy buildings* (*nZEB*). In 2013, the EU decided to oblige all member states to achieve NZEB by 2020 [28]. Under the pressure of the EU, several countries started to prepare their construction sectors for this transition but were challenged with several technical, financial, and social barriers [7]. Therefore, several calls were made to adopt nZEB who are less performing than NZEB. A nZEB is considered to be one building with a low energy demand, between 15 and 30 kWh/m$^2$/year for heating and cooling energy needs, where 30% of demand is supplied with renewables [7,29].

Another important concept that emerged in the last five years is the concept of *Healthy Net Zero Energy Buildings*. Several countries that implemented the NZEB approach via the Passive House standard faced serious indoor environmental quality (IEQ) issues [1]. To guarantee the user's comfort and, at the same time, to reach the goal of a zero-energy balance, the healthy NZEB concept emerged. Some of the most common problems seen in NZEB are overheating in summer due to the high level of airtightness, insulation, and the overestimation of the passive cooling potential [30–32]. The unfamiliarity with the use of mechanical systems for ventilation in NZEB households resulted in poor air quality problems and high concentration of carbon dioxide. In this sense, the design of healthy NZEB, as well as revising the definition of NZEB, must consider a minimum IEQ requirement, focused on assuring air quality, suitable amounts of natural light, shading systems to avoid overheating, and a design of control systems that considers the technical capacity of the occupants.

Learning from the evolution of the NZEB definition in Europe provides valuable insights. Therefore, a new NZEB definition in Chile should address the highlighted concerned discussion above.

## 3.2. NZEB Principles

Based on our literature review, we identified four principles to design NZEB. The four principles address energy efficiency, indoor environmental quality, renewable energy, and carbon emission associated with energy consumption. Table 2 lists series of measures that can be applied for NZEBs. Depending on the climate and building type, designers can use those principles to design and operate NZEB. The principles implicate to identify the best fit to purpose measures during the design initiation phase:

(1)  Reduce the energy demand for all newly constructed buildings. The energy demand value is for the sum of the demands of buildings, space heating, space cooling, DHW, auxiliary energy, ventilation, lighting, and appliances [1].

(2)  Improve IEQ, allowing for maximum thermal comfort and avoidance of overheating. This includes air quality control through mechanical ventilation [1].

(3)  Fix a percentage of renewable energy demand to be covered by renewable energy annual balance.

(4)  Reduce the overarching value for primary energy consumption and carbon emissions per year. It is also important to amend additional measures to address mobility and materials' embodied energy issues [1].

**Table 2.** The four principles of NZEB design [1].

| First NZEB Principle | Second NZEB Principle | Third NZEB Principle | Fourth NZEB Principle |
|---|---|---|---|
| Reduce energy demand | Improve indoor environmental quality | Provide renewable energy share | Reduce primary energy and carbon emissions |
| • Reduce internal loads<br>• Reduce building envelope loads<br>• Reduce HVAC equipment energy consumption | • Set up minimum fresh air per person<br>• Enable Natural Lighting<br>• Set up a maximum occupant density | • Produce energy from renewable sources onsite<br>• Introduce renewable energy delivered from nearby or offsite<br>• Avoid double counting | • Reduce the primary energy demand<br>• Reduce the carbon emissions related to delivered energy |

For NZEB, bioclimatic design and energy efficiency are the first step to determine in relation to thermal comfort. Building performance simulation tools must be used to assess the comfort conditions,

air exchange volumes, and the energy balance of renewable energy systems. Design reviews, third party commissioning, and continuous monitoring and adjustments will be the assurance of quality performance of NZEB.

### 3.3. A Framework for a New Standard for Chile

As can be found in the previous section, there are several definitions and principles that we respect and applied, in one way or another, to come up with a framework for a new NZEB standard for Chile. However, we should identify NZEB in a different way than Europe to achieve low-tech NZEB. Table 3 provides a comparison between the different technological approaches to reach NZEB. Also, Figure 3 provides an illustration of the new standard framework and the suggested two approaches to assess the performance of NZEB in Chile. Based on our literature review and context analysis, we identified three main components that must be addressed in any future NZEB standard in Chile:

**Table 3.** Comparison between the high-tech and low-tech approach for NZEB.

|  | High-Tech Approach | Low-Tech Approach |
|---|---|---|
| **Energy Efficiency Target:** | 15–25 kWh/m$^2$y | min. 15–45 kWh/m$^2$y |
| **Renewable Energy Target:** | 15 kWh/m$^2$y | 30–45 kWh/m$^2$y |
| **Envelope:** | Max. Insulation and air tightness static and adaptive models | Max. bioclimatic and passive design solutions |
| **Thermal Comfort:** | Static and adaptive models | Adaptive models |
| **Air Quality:** | 800 ppm | 1200 ppm |
| **Behavior:** | Conscious and based on rigid operation schedules | Conscious and adaptive |
| **Systems:** | Mechanical ventilation with heat recovery, ultra-efficient HVAC systems. | Hybrid ventilation, with individual heating and cooling unit. |
| **Controls:** | Building Management Systems (BMS) | None or manual |
| **Monitoring:** | Real-time and full monitoring using smart meters. | Monthly and manual energy consumption readings |
| **Operation:** | Full time dedicated expert or facility manager | By users |
| **Cost:** | Cost-Optimality calculation every 5 years coupled to incentives | Cost-Optimality calculation every 5 years coupled to incentives |

The first component must address thermal comfort. In industrial countries, thermal comfort is well-established. The building services industry reached almost a 100% penetration in all newly built and existing buildings. The high-tech definition of NZEB is based on stringent comfort models [1]. As shown in Figure 4, a stringent comfort model like the static comfort model of ASHRAE (yellow and purple lines) means that temperatures should remain between a narrow range of temperatures and consequently, result in high energy consumption. However, a low-tech NZEB definition does not need to be based on stringent comfort models that do not consider the outdoor temperatures. There is a serious opportunity to develop new definitions and concepts for NZEB in non-industrial countries that integrate more tolerant adaptive comfort models, reflecting the socio-economic status [1,30,33]. The use of adaptive comfort models for NZEB can significantly reduce energy consumption and make it easier to achieve annual energy neutrality. Literature indicates that an over focus on energy performance can lead to health problems and discomfort [34]. For example, occupants might sacrifice their thermal comfort to achieve low energy consumption [34].

The component is related active systems. By "active systems" we mean building energy systems such as mechanical ventilation systems, heat exchangers, and cooling and heating systems. This component also addresses building management systems. As shown in Table 3, a high-tech NZEB relies on mechanical ventilation with heat recovery. Those systems and building services installations are most of the time imported and coupled with sophisticated building controls. Therefore, we think that Chile should not necessarily follow the European Passive House Label or the Active House Label [35,36] that require high-tech equipment. Depending on the local climate and the technological infrastructure of Chile, a set of low-tech building service products and solutions can cater towards a low-tech NZEB. The use of ceiling fans or movable heaters with gas canister in NZEB can be effective examples to avoid costly hydronic central heating systems. There is an opportunity to develop new

building service technologies such as heat pumps, which are adapted to the socio-economic status, human physiology, and local energy market in non-industrial countries [1].

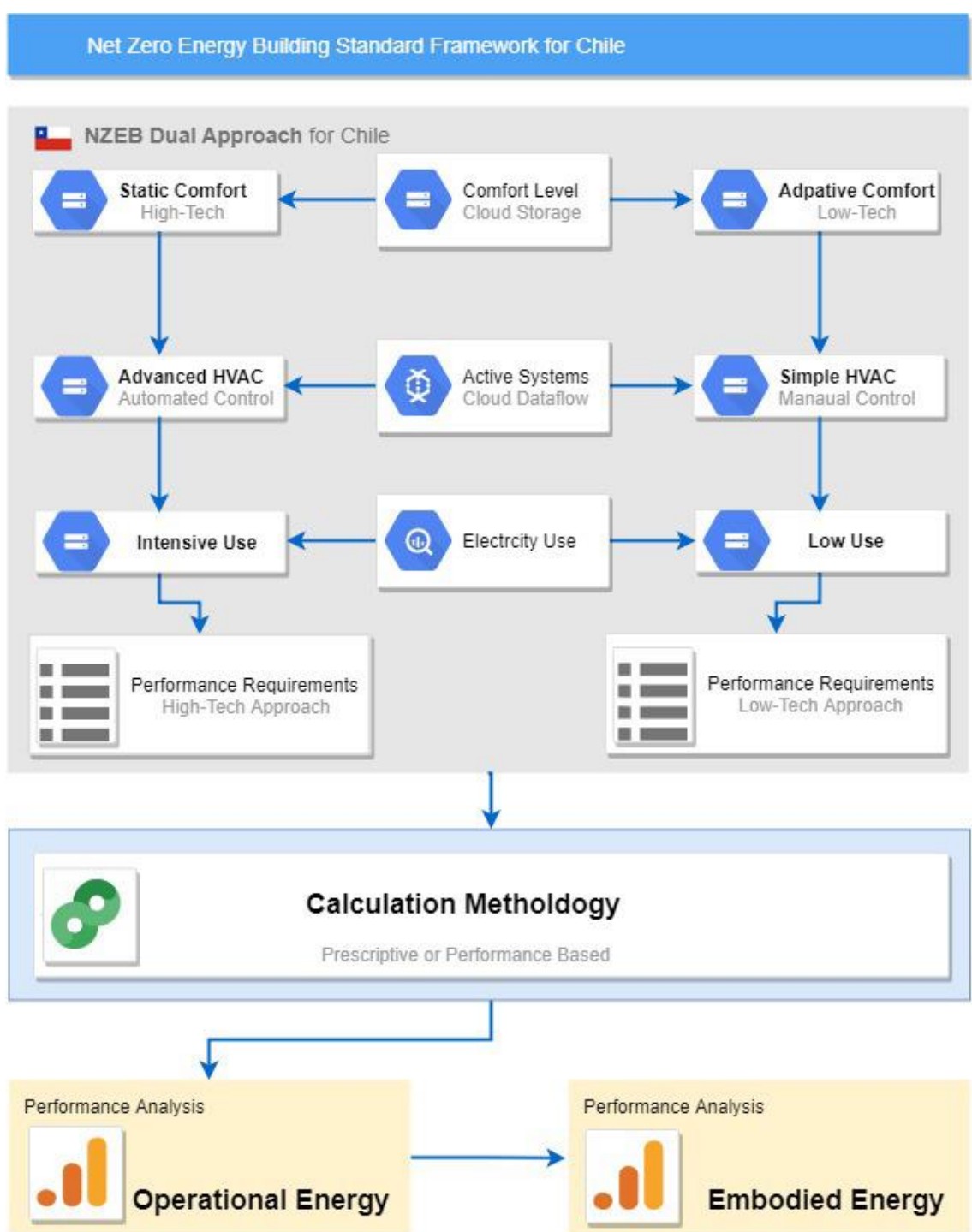

**Figure 3.** The dual approach of proposed NZEB standard framework for Chile.

The third components address electrification and impact of appliances and lighting use on the total energy consumption. In high-tech NZEB, appliances dominate energy use, and energy breakdown of total consumption shows an insignificant contribution of heating or cooling energy needs to the total energy consumption. NZEB have very low energy needs associated with cooling and heating. The highly insulated envelopes and airtight spaces shift the consumption focus towards appliances

and plugs. With the proliferation of electronic devices and penetration of ultra-efficient lighting units, NZEB shift from thermal dominated loads to electric dominated loads. Therefore, the potential of electrification of NZEB in non-industrial countries is promising. The use of heat-pumps and ultra-efficient electric appliances can result in very low energy use intensity, making the net-zero target much more achievable in non-industrial countries [1,7].

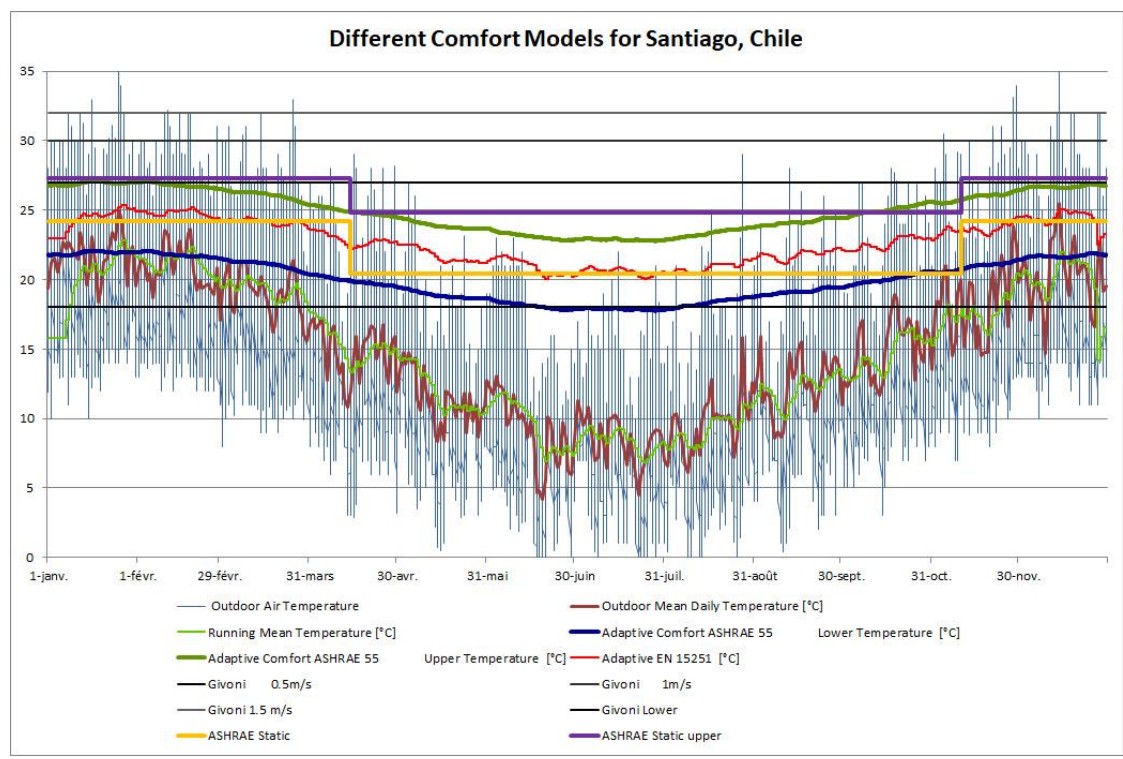

**Figure 4.** A comparison of different comfort models for the climate of Santiago.

Finally, we proposed a calculation method for the proposed new NZEB standard that considers the decarbonization as shown in Figure 5, based on the work of Attia [1]. The advantage of calculating the net zero source energy (primary energy) is that it allows for quantifying the carbon emissions associated with NZEB operation at the source (utilities), which is beneficial on the national level. A source zero energy building means producing the same amount of the consumed energy measured at the utility source. This means that the losses from conversion and transmission in both directions are considered. The source energy is the primary energy at the source of energy production. To estimate the building total source energy, the imported and exported energy is multiplied by the appropriate site to source conversion factor. Primary energy is calculated from delivered and exported energy as:

$$PE = \frac{PE_{,nren}}{A_{net}}$$
$$PE_{nren} = \sum_{i}(E_{deliv,i}, f_{deliv,nren,i}) - \sum_{i}(E_{exp,i}, f_{exp,nren,i})$$

(1)

where

$PE$ is the primary energy indicator (kWh/m$^2$.a);
$A_{net}$ is the net used floor area (m$^2$) calculated according to national definition;
$PE_{nren}$ is the non-renewable primary energy (kWh/a);
$E_{deliv,i}$ is the delivered energy on site (kWh/a) for energy carrier $i$;
$f_{deliv,nren,i}$ is the non-renewable primary energy factor for the delivered energy carrier $i$;
$E_{exp,i}$ is the exported energy on site (kWh/a) for energy carrier $i$;

$f_{exp,nren,i}$ is the non-renewable primary energy factor for the delivered energy compensated by the exported energy for energy carrier *i*; which is equal to the factor of the delivered energy, if not defined nationally in different way.

Consequently, the carbon emissions associated with the energy use can be calculated based on the delivered and exported energy.

$$PE_{CO2} = \frac{\text{Emission } CO2}{A_{net}} = \frac{\sum_i \left( E_{deliv,i} CE_{deliv,i} \right) - \sum_i \left( E_{exp,i} CE_{exp,i} \right)}{A_{net}} \tag{2}$$

where

$PE_{CO2}$ is the $CO_2$ emission indicator ($kgCO^2/m^2$.a);

Emission $CO2$ is $CO_2$ emission ($kgCO_2/a$);

$A_{net}$ is the net used floor area ($m^2$) calculated according to national definition;

$E_{deliv,i}$ is the delivered energy on site (kWh/a) for energy carrier *i*;

$CE_{deliv,i}$ is the $CO_2$ emission coefficient ($kgCO_2/kWh$) for the delivered energy carrier *i*;

$E_{exp,i}$ is the exported energy on site (kWh/a) for energy carrier *i*;

$CE_{exp,i}$ is the $CO_2$ emission coefficient ($kgCO_2/kWh$) for the exported energy carrier *I, depending on national definition*.

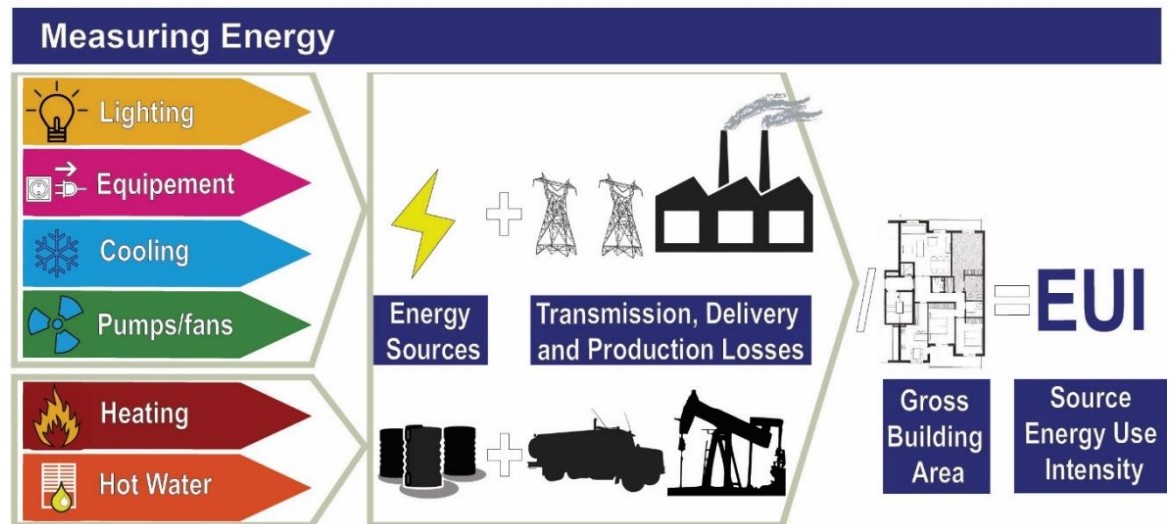

**Figure 5.** Graphical equation for the primary energy use breakdown and intensity for net zero energy balance calculation [1].

*3.4. Focus Group Discussions*

The focus groups full report can be found in Appendix A under Tables A1–A5. Key ideas are highlighted under the following statements:

(1) The absence of environmental taxation systems for fossil energy and energy pricing policy, results in economic subsidies for investments in buildings energy efficiency.

(2) NZEB green technologies investment costs are viewed as excessively high particularly in relation the high real estate cost and low wages in Chile. A fact that stand as a barrier in the proliferation of NZEB [34].

(3) The lack of institutional infrastructure to enforce the compliance with the TDRe and building certification and the absence of penalties for code violation.

(4) The current building code (TDRe) is outdated and need to align with the NZEB approach.

(5)　There is a lack of professional qualification and vocational education and technical staff ready to handle NZEB solutions and technologies.

(6)　Public awareness of the end-users on the importance of healthy and climate neural buildings is limited.

(7)　There is an underdeveloped market and absence of an industrial infrastructure to carry on an ecosystems' of NZEB components and materials producers and suppliers. The Chilean market is open to imports without protection and investments in local made energy efficiency technologies.

## 4. Discussion

### *4.1. Summary of Main Findings*

For this study, we developed a framework for a new standard for NZEB that can be used by policy makers in Chile and Latin America. By mapping definitions and performance standards and indicators of NZEB in Europe, the proposed framework seeks to identify and articulate the crucial performance of NZEB in Chile. The proposed framework, in Figure 3, is a result of the literature review analysis and focus groups discussions. The framework offers a dual approach to reach a new NZEB standard in Chile. The framework highlights the technical, societal, and economic concerns of experts regarding the feasibility of implementing NZEB in Chile and involves thermal comfort, active systems, and internal loads as well a primary energy calculation method as major components.

This framework was based on the identification of different definitions and performance criteria, and the focus group discussion with more than 60 Chilean and European experts. One of the deep-rooted problems of NZEB in Chile is not their additional cost associated with their construction. The root-cause problem is the lack of industrial-manufacturing infrastructure that allows firms to produce capital NZEB products and equipment (e.g., heat exchangers, heat pumps, photovoltaic panels, insulation materials, etc.). The weak legal framework for comfort and energy efficiency in the building sectors and the subsidized energy pricing policy, decrease the chance to create a market demand for NZEB. The integration of local and advanced building technologies into NZEB is the first challenge in the construction sector. However, we proved that there is a need to create a context-specific standard for NZEB. In general, there is an underestimation of the important role of setting an NZEB standard to comply with the OECD requirements to achieve comfort and energy savings, and to curb the carbon emission associated with the building sector. Finally, participants in the focus groups discussions agreed that Chile is not yet ready to cross those barriers today; however, NZEB remain a promising concept that is under early development.

Based on our literature review and focus groups discussions, we can confirm that there is a serious need to address the following criteria to develop a new standard for NZEB in Chile:

(1)　The identification of context-specific comfort requirements for all building types to allow setting the performance thresholds for NZEB. Comfort definition and fuel poverty are an important challenge in Chile [33] that need to be addressed first.

(2)　Developing and adopting a low-tech alternative of the Passive House Standard is a crucial approach to define a low-tech NZEB. There is a need to develop new building service technologies that are adapted to the socio-economic status and local energy market in Chile [1].

(3)　We developed and validated a set of performance indicators; metrics and performance threshold for a new NZEB standard in Chile (see Table 3).

(4)　Reduction of plug-loads and lighting can facilitate achieving NZEB. The need for building decarbonization will bring electrification to Chile. Therefore, any new standard should address electric loads and suggest measurable consumption limits and foresee the grid flexibility [7].

(5)　We developed and validated a calculation method for the energy balance that should be included in the new NZEB standard in Chile (see Equations (1) and (2)).

The role of NZEB is crucial to improve comfort, outdoor air quality, and reduce carbon emissions. It is vital for Chile to develop a new standard that addresses the criteria above and to prepare for an integrated policy to promote NZEB in the Chilean building sector.

*4.2. Strengths and Limitations of the Study*

We are not aware of any conducted study that aimed to set up a framework to create new standards for NZEB in Chile or Latin America. Despite the challenges of creating a consensus-based standard for NZEB in Chile, the research benefited from the contributions of European and Chilean experts. The two focus group discussions fostered a consensus-based discussion to create a novel framework that serves as a basis for a new standard. The research aimed to provide an analysis of the existing literature and body of knowledge in order to define an integrated policy for energy efficiency represented in the form of a standard framework.

The methodology used in the study was based on literature review and focus group discussions. The present study's approach remains novel in because it involves locals and valorizes their expertise and combines this expertise with learned lessons from Europe. This combination allowed qualitative evaluation of the technology maturation and barriers of market adoption of NZEB in Chile in relation to users' comfort.

We proposed and validated framework for a NZEB standard within the scope of Chilean and Belgian cooperation project "Resilience and Net Zero in Buildings" activities. This framework identified KPIs that should be selected to design and evaluate NZEB's performance during the construction and operation phase while empowering users. The developed framework and key criteria identified in the study will improve the understanding of policy makers, and allow for comparison, discussion, and learning. In other words, it will allow benchmarking of NZEB performance, so that researchers can measure their social, economic, and environmental sustainability [1].

The two focus group seminars confirmed our findings and highlighted the importance of bringing NZEB into mainstream construction.

It is acknowledged that capital expenses and initial costs play a major role in decision-making processes regarding refurbishment and construction of new buildings in Chile. These economic considerations also have a direct influence on technical performance characteristics of NZEB and are therefore important to consider in the context of the present study. We nevertheless decided to discuss these aspects only briefly, because the identified research gap outlined the need for addressing various barriers concerning performance assessment of technological aspects first. We expect that the economic studies will follow suit once the technical performance metrics and evaluation procedures of NZEB get consolidated and suggest that future studies focus on ways of reconciling NZEB performance, with options for profitable business operations for both the client and the real estate industry [1].

Another shortcoming of our study is failing to address the embodied energy. Our definition and framework focused mainly on the operation energy and did not address the building performance during different life cycles. There is an increasing awareness of the importance of embodied energy of high-performance buildings, and several studies indicate that NZEB are just half the plan to tackle climate change [37,38]. The challenge of decarbonization is technological and economic and should include thinking about building materials. As mentioned earlier in Section 3.2 it is also important to amend additional measures to address mobility and materials' embodied energy issues for NZEB [1,39].

*4.3. Implications for Practice and Future Research*

NZEB are not an option that Chile can avoid. Chile is considered a developed country and has been classified as a middle-income country according to the World Bank. The construction market registered an average growth rate of 4% between 2010 and 2020 in Chile. Thus, the market is expected to double this growth rate in the coming 10 years. Currently, Chile possess 6.5 Million households that are expected to add 0.5 Million household until 2030. At the same time, the Chilean government is introducing a range of measures to expand its use of renewable sources and increase energy efficiency

of the building sector. In this context, NZEB are an opportunity to achieve occupant well-being, and achieve the national carbon emission reduction target. In this study, we initially identified and classified KPIs to assess NZEB. We are expecting that our framework will get translated into a novel standard, and in the long-term increase NZEB's market penetration in Chile and Latin America. Based on this study, an initial framework for new NZEB standard is presented. Like the definitions and standards developed in the EU member states and OECD countries, Chile must develop a new standard for NZEB evaluation that empowers users. Our framework suggests a multi-criteria framework that groups most parameters under three KPI categories. We find it important that future research builds on our findings and develops more consistent standards and evaluation frameworks for NZEB. Finally, the present study is the first Latin American milestone widening the research about NZEB definition. Owners, contractors, energy companies, regulators, local officials, and building professionals all will play a major role in the market transition toward NZEB. Therefore, it is crucial to develop an integrated policy for NZEB in Chile.

## 5. Conclusions

NZEB is one of the most important concepts in the construction industry through which energy-saving potentials and low carbon emissions can be achieved. They are of principal importance not only for industrial countries, but also non industrial countries. The Chilean government is well grouped and organized around the themes of climate resilience, affordable housing, and energy efficiency. However, the construction sector and real estate market need to adapt rapidly to digest the advanced concept of NZEB and operate in an optimal way that can reduce (operational and embodied) buildings' environmental impact and empower users. To be able to communicate the potential advantages of NZEB, we developed a framework for a new NZEB standard which provides holistic performance criteria of NZEB. The technical feasibility of the framework has been validated in focus groups. With the framework, the authors open the scope of NZEB and link it to building scale, environmental performance, buildings' decarbonization and occupancy centered fields. It is expected that the framework helps policy makers to create a new definition for NZEB in Chile and shape a new energy policy in the building sector.

The framework identifies and classifies the large variety of performance indicators for NZEB, thereby starting a process toward the development of a new Chilean Standard for high performance buildings. The framework is not only useful for future research development but is also needed for immediate practical purposes. Chile has an opportunity to learn from the pitfalls of NZEB implantation in Europe. Buildings are expected to become more tightly bound to digital technologies and advanced building systems, and at the same time, decarbonized. The industrial integration capabilities and strength in Latin America will depend on addressing the NZEB and creating new frameworks for new standards. Using the proposed framework, a new standard within an integrated policy can be better developed to benchmark NZEB in Chile.

**Author Contributions:** Conceptualization, M.B.P. and S.A.; Data curation, M.B.P.; Funding acquisition, M.B.P. and G.v.M.; Methodology, M.B.P. and S.A.; Project administration, M.B.P. and G.v.M.; Supervision, M.B.P. and S.A.; Validation, G.v.M. and S.A.; Visualization, F.V. and S.A.; Writing—original draft, F.V.; Writing—review & editing, M.B.P. and S.A.

**Funding:** This paper has been developed in the context of the bilateral cooperation project between Chile and Wallonia-Brussels federation called "Resilience and Net Zero in Buildings", with the cooperation of the Sustainable Architecture and Construction Research Group of University of Bio-Bio, Chile. The APC was funded by the University of Bio-Bio in Chile.

**Acknowledgments:** This paper has been developed in the context of the bilateral cooperation project between Chile and Wallonia-Brussels federation called "Resilience and Net Zero in Buildings", with the cooperation of the Sustainable Architecture and Construction Research Group of University of Bio-Bio, Chile. Special thanks to Muriel Díaz Cisternas and Laura Marín Restrepo.

**Conflicts of Interest:** The authors declare no conflict of interest.

## Appendix A

The focus group discussion symposium 2 was held on Thursday, 27 July in the auditorium of the faculty of engineering in wood, under the modality of discussion tables. The lecturers and experts in the subject were distributed at the tables, each of one with an assigned topic and with a question to guide the discussion. Likewise, a representative was chosen to record the most important ideas and consolidating the response of each table, which was presented at the end of the event.

Here is the summary of the most important ideas presented by each table (see Tables A1–A5).

**Table A1.** Technology and Architecture Zero Energy.

| |
|---|
| What technology to develop or transform for the implementation of Zero Energy buildings in Chile? Experts: Shady Attia \| Andrés Montero \| María Elena Soldatti Representative: Laura Marín Restrepo |
| There are technologies in Chile that can support the implementation of Zero Energy buildings, as well as the potential for the use of alternative energies and the technical knowledge for its development. Therefore, the implementation challenges are political and cultural, as well as it is essential to educate the end users of the buildings on issues of sustainability and efficiency. Before thinking about technologies or specific design, the city must be planned. Having a regulated and planned urban growth is necessary for energy efficiency and environmental comfort strategies can work. Likewise, it is important to optimize passive strategies first, since in some regions in Chile it is not necessary to invest heavily in technology so that a building consumes less energy and people are comfortable. It is also considered that culture is important within the space needs and comfort requirements, so it does not apply the same standard. The concern is generated, for example, how to conserve heat and/or heating without using mechanical ventilation? Is it possible? The Zero Energy standard points to greater isolation and necessarily involves mechanical ventilation, however, people in Chile, or at least Concepción, are not familiar with airtight and automated spaces. Can Chileans adapt to artificial environments? Is it necessary in this context? It reflects that a balance must be found between the perception of people and their requirements, with the mechanical systems that are implemented. The role of real estate and state regulation is fundamental since they usually point to interests that differ. There is pressure from real estate and the construction market to reduce energy efficiency requirements. The option of centralized heating systems, for example, should be stated initiatives, because the real estate market does not support it because it is not convenient for it. Remains the concern about how communities would use these systems in a society such as Chile. On the other hand, mandatory certification is necessary for effective communication with users. If the buildings declare their performance and the users are informed, they will know what decisions to make and could demand efficient buildings. Similarly, given that there is a gap between the design/technologies and the construction and operation of the buildings, it is necessary to invest in skilled labor and in trained users to make an actual implementation. Regarding users, strategies should be sought to change the mentality of making decisions based on immediate impacts by decisions based on performance and long-term, to make investments. For this, the technology must be durable, because, in that way, the users know that the investment is worth it. Likewise, buildings must be functional, fulfil their basic purpose and be simple, so that people accept and demand Zero Energy buildings. Technologies do not need to be high range, they can be simple, and they are already in the market. Simple home automation, for example, can be used to support efficient use. Finally, regarding the heating requirement, which is the one that demands the most energy in the southern of Chile, it is believed that a change of mentality in people must be pointed out, because, although there are more efficient and less polluting technologies, traditional heating is cheaper, and people do not want to change it. |

**Table A2.** Zero Energy Retrofit.

What opportunities or restrictions has the application of the concept of Zero Energy Architecture in buildings retrofit in Chile?
Experts: Luis Braganza | Geoffrey Van Moeseke
Representative: Jeremy Piggot

**Opportunities**

Cultural/economic

- Incorporating the concept of Zero Energy Architecture in buildings retrofit allows improving the hygrothermal, luminous and acoustic comfort for users.
- Long-term energy savings generates significant monetary benefits.
- The incorporation of this type of measures in the building helps to achieve the energy independence in much of the building of the country.
- It generates a synergy that allows the production of jobs in areas already consolidated for professionals and construction workers.
- If this type of rehabilitation is tackled collectively the benefit may become more feasible.

Urban

- Urban Regeneration and Smart Cities. The city can generate a brand associated with the rehabilitation of its older infrastructures.
- The urban centers have suffered a decay and the urban expansion for residential purposes occurs mainly in the periphery. The old buildings located in the center of the city offer the opportunity to be rehabilitated and made more attractive if they integrate bioclimatic variables.

Environmental

- The reuse of structures decreases carbon footprint.

**Restrictions**

Cultural/economic

- A change in the mentality of the people is necessary since retrofitting is not usually quite striking because although economic compensation can be quite beneficial in long-term, the initial costs associated can be very high.
- Retrofitting buildings with these criteria need instructed users to make a correct use of this.

Governmental

- Lack of legislation and financial assistance from the government, the incorporation of rehabilitation subsidies with Zero Energy Architecture criteria could encourage the reduction of a large part of the demands on existing buildings in Chile.

Retrofitting with Zero Energy criteria in heritage buildings can become quite complicated due to the legal restrictions that exist regarding its modification, and the impact on the aspirations of the community.

**Table A3.** Architecture Zero Energy Politics.

---

What opportunities or restrictions has the application of the concept of Zero Energy Architecture in new constructions in Chile?
Experts: Felipe Encinas | Javier del Rio
Representative: Susan Agurto

---

It is agreed that in Chile is not feasible to implement the Net Zero standard, but what must be done from the scope of the policy is to define its own standard, feasible to implement and according to the Chilean context.

The economic factor, investment and cost of the constructions prevent that a standard like the Net Zero can be implemented. It is necessary to differentiate the solutions considering the conditions of local comfort and not under wide parameters that do not differentiate the exigency according to the climate of each locality.

It is necessary to implement economic incentives that motivate the private to invest in such constructions and at the same time create a system that regulates the value that the private will estimate in their projects since the objective would be that the costs for the users do not increase. The normative change is necessary and complementary to the energetic qualification and certification. Progress must be made in both aspects.

It will be necessary, in addition, to carry out an important diffusion campaign to put in value the implementation of sustainable constructions, creating this awareness in the users.

It is necessary to develop a clear roadmap: to update thermal regulation so that it is a requirement to the private; to consolidate the information in a clear plan and to be a legal instrument, as well as to make the population aware of the concepts of sustainability.

---

**Table A4.** Comfort, Energy Poverty and Zero Energy Architecture.

---

What challenges has the implementation of Zero Energy Architecture to achieve environmental comfort in buildings and reducing energy poverty in Chile?
Experts: Cristina Engel
Representative: Paulina Wegertseder

---

The measures to be implemented must respond to the context in which it is inserted. Perhaps it is easy to reach a Zero Energy standard, but is it enough to achieve it? Are there no other priority problems? Is it what people need?

There are user habits that show that they are not prepared for certain strategies, for example, ventilation. A challenge is to be part of the design process to the user because they will realize its role. Involving the user is key for a correct use of the building and greater awareness.

In general, independent of the building sector that seeks to improve, it is necessary to involve the user from two perspectives: 1. In the design process as a fundamental variable, as well in the construction stage; 2. Educate him for the stage of use of the building.

On the other hand, it is not enough to reach a Zero Energy standard in a building if you do not consider other deficiencies that influence the user comfort. Energy efficiency must be integral, as a strategy or goal.

---

**Table A5.** Zero Energy Architecture Implementation.

---

What actions should be taken to ensure that the community demands Zero Energy buildings in Chile?
Experts: Beatriz Piderit | Jesús Pulido
Representative: Matías Tapia

---

**Detecting** key stakeholders (individuals and institutions) across the whole society, influencing decision-making in the construction sector.
**Organizing** activities that integrate these actors, communities and private and public institutions, with all intermediate levels needed to be carried out (broadcasting, meetings, conversations). Efforts should be made to include the private sector (real estate, industrial sector, etc.).
**Conducting** informational workshops to communities, for example, neighborhood associations, social groups or parent centres, to address the issues of energy efficiency of buildings. Participation of parents in school communities can be used to make broadcasts.
Consolidation report: Laura Marín Restrepo
English version 1: 21 August 2017

---

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
