# Peer review of "Net Zero Buildings—A Framework for an Integrated Policy in Chile"

_sustainability, doi:10.3390/su11051494_

Round 1

Reviewer 1 Report

It is considered that the article has some important weaknesses that should be deeply revised for possible publication. Here are explained in detail:

- Introduction

There is a lack of reference in some parte of this section. It is important to give adequate literature which sustains your arguments:

Page 3. Lines 77 to 81.

Page 3. Second paragraph.

- Methodology

In my opinion, this article suffers from a methodology that is not too rigorous, because it has been presented in a very superficial way.

The methodology for the literature review has to be described in detail. From which database did you obtain documents? What was your way of finding the most relevant literature in the field? Are you sure that the used reference are the most significant and recent?

It is necessary the description of the Focus Group Discussions and how you carried out them. To ensure the quality of results from FGD is necessary define the structure of sessions, the adequate composition of each groups (level of expertise, gender, nationality, number of participants, etc); and also the information given previously and during the session.

You can get some ideas from this paper:

Sierra-Pérez, J., López-Forniés, I., Boschmonart-Rives, J., & Gabarrell, X. (2016). Introducing eco-ideation and creativity techniques to increase and diversify the applications of eco-materials: The case of cork in the building sector. Journal of cleaner production137, 606-616.

- Results and discussion

A recent significant issue in this type of buildings is the amount of embodied energy in their construction, which begins to be relevant with the operational energy of their energy life cycle. It is important to incorporate. It is important to incorporate this topic into the discussion from the beginning, since by reducing the energy used during the life of the building, the energy you use for its construction, maintenance or elimination becomes more relevant:

F. Pacheco-Torgal, J. Faria, S. Jalali. Embodied energy versus operational energy. Showing the Shortcomings of the Energy Performance Building Directive (EPBD)

Mater Sci Forum (2012), 

Sierra-Pérez, J., Rodríguez-Soria, B., Boschmonart-Rives, J., & Gabarrell, X. (2018). Integrated life cycle assessment and thermodynamic simulation of a public building’s envelope renovation: Conventional vs. Passivhaus proposal. Applied Energy212, 1510-1521.

In general, the document's findings are not surprising, as it was quite expected due to NZEB's own definition. Therefore, it would be important to incorporate other issues, such as the issue of the energetic implications of the construction of this type of buildings. These relevant issues may come from a more in-depth literature review.

Author Response

-Reviewer 1

It is considered that the article has some important weaknesses that should be deeply revised for possible publication. Here are explained in detail:

Thank you very much Reviewer 1. We appreciate your time and effort and we are very happy to receive such a constructive feedback. We did our best to address all your comments. Please follow our modifications explained point by point.

- Introduction

There is a lack of reference in some parts of this section. It is important to give adequate literature which sustains your arguments:

Page 3. Lines 77 to 81.

Page 3. Second paragraph.

We agree with reviewer one. We addressed this comment in the text. We added four more recent references that support our claim in this section. Changes can be found in page 3.

- Methodology

In my opinion, this article suffers from a methodology that is not too rigorous, because it has been presented in a very superficial way. The methodology for the literature review has to be described in detail. From which database did you obtain documents? What was your way of finding the most relevant literature in the field? Are you sure that the used reference are the most significant and recent?

It is necessary the description of the Focus Group Discussions and how you carried out them. To ensure the quality of results from FGD is necessary define the structure of sessions, the adequate composition of each groups (level of expertise, gender, nationality, number of participants, etc); and also the information given previously and during the session.

You can get some ideas from this paper:

Sierra-Pérez, J., López-Forniés, I., Boschmonart-Rives, J., & Gabarrell, X. (2016). Introducing eco-ideation and creativity techniques to increase and

diversify the applications of eco-materials: The case of cork in the building sector. Journal of cleaner production, 137, 606-616.

We agree with reviewer one. We appreciate the constructive feedback and benefited from reading the cited work. We addressed this comment in the text. We performed a major modification and re-wrote this whole section. We mentioned the databases that were consulted. We amended the focus group discussions description with detailed information regarding the data collection process and data quality. Also, we added the focus group discussion report in Appendix A. Changes can be found in Section 2.

- Results and discussion

A recent significant issue in this type of buildings is the amount of embodied energy in their construction, which begins to be relevant with the operational energy of their energy life cycle. It is important to incorporate. It is important to incorporate this topic into the discussion from the beginning, since by reducing the energy used during the life of the building, the energy you use for its construction, maintenance or elimination becomes more relevant:

F. Pacheco-Torgal, J. Faria, S. Jalali. Embodied energy versus operational energy. Showing the Shortcomings of the Energy Performance Building Directive (EPBD) Mater Sci Forum (2012),

Sierra-Pérez, J., Rodríguez-Soria, B., Boschmonart-Rives, J., & Gabarrell, X. (2018). Integrated life cycle assessment and thermodynamic simulation of a

public building’s envelope renovation: Conventional vs. Passivhaus proposal. Applied Energy, 212, 1510-1521.

In general, the document's findings are not surprising, as it was quite expected due to NZEB's own definition. Therefore, it would be important to incorporate other issues, such as the issue of the energetic implications of the construction of this type of buildings. These relevant issues may come from a more in-depth literature review.

We agree with reviewer one. We appreciate the constructive feedback and benefited from reading the cited work. We addressed this comment in the text. We performed a major modification and re-wrote this whole section. We mentioned articulated the importance of embodied energy. Also, we rearticulated the major findings and contribution of the research and supported the documents with two new figures. Changes can be found in Section 3.

Reviewer 2 Report

Overall, this paper reads more like a consultancy project report. Although the arguments and the findings presented are plausible, the rigor of research design is questionable. The literature review part is done in a highly broad-brush manner and limited. The quality of the literature survey and review is low, with only a very limited number of journal articles covered (which also includes self-citations). A structured and systematic literature reivew is needed. For the interview part, the methods for question design, data collection and result analysis apear casual and sketchy. There is lack of a structured framework to provide a coherent connection between the two parts, i.e. the literature and the interviews. Also, the rationale for using such research design is not strongly supported. 

In sum, this paper suffers many gaps and cannot be recommended for publication before making some substantial improvements, including some further work to enhance its research design and methodology.

Author Response

-Reviewer 2

Overall, this paper reads more like a consultancy project report. Although the arguments and the findings presented are plausible, the rigor of research design is questionable. The literature review part is done in a highly broadbrush manner and limited. The quality of the literature survey and review is low, with only a very limited number of journal articles covered (which also includes self-citations). A structured and systematic literature reivew is needed. For the interview part, the methods for question design, data collection and result analysis apear casual and sketchy. There is lack of a structured framework to provide a coherent connection between the two parts, i.e. the literature and the interviews. Also, the rationale for using such research design is not strongly supported.

In sum, this paper suffers many gaps and cannot be recommended for publication before making some substantial improvements, including some further work to enhance its research design and methodology.

Thank you very much Reviewer 2. We appreciate your time and effort and we are very happy to receive such a constructive feedback. We did our best to address all your comments. Please follow our modifications explained point by point

We addressed the comment regarding the literature review limitation. We performed a major modification and added more than 12 new references. We mentioned the databases that were consulted to perform the literature review. We could not avoid self-citation because the one of the main authors in specialised in net zero energy buildings and study build up on previous knowledge. However, we made sure that the self-citation does not exceed more than 7 references after deleting one reference. Changes can be found in Section 1.

We addressed the comment regarding the methodology rigor. We amended the focus group discussions description with detailed information regarding the data collection process and data quality. We indicated the background, structure, procedure and quality assurance measures of the focus group discussion. Also, we added the focus group discussion report in Appendix A. Changes can be found in Section 2.

Thank you, reviewer, 2. We are open for any further specific improvement.

Reviewer 3 Report

General

The paper address an interesting topic, perhaps especially so given the nature of urbanisation and building. At the moment, however, the context is missing, and the paper suffers from a lack of detail.

To that end, please discuss the findings more. The paper would read better with more elaboration of key and contentious points. The narrative for why a standard is needed, how the methodology helps in this quest, and what the framework is and does vis-a-vis health and energy issues needs to be strengthened. The paper would also gain from being more direct, e.g. instead of "This paper will show that a framework for NZEB in Chile is necessary...", try "A framework for NZEB in Chile is necessary..." 

English Language and Style:

Abstract needs to be thoroughly checked for capitalisation and punctuation, e.g. at the beginning of a sentence, for acronyms and proper names, and for chemical symbols (e.g. CO2). Other sections require a proofread for spelling and punctuation (e.g. double period in line 217). Could the acronyms be limited to heavily used ones in the paper? For example, FGD could simply be "focus group discussion" as it is used in only a few places. Limiting the acronyms would make for a more readable paper.

Please ensure the Bibliography follows the Journal style for italicisation, punctuation and capitalisation.

Introduction:

Table 1 should align with the discussion in the text. For example, the text discusses France, but France is not in Table 1. The text discusses auxiliary energy and heating and cooling energy, yet Table 1 does not break it down that way. Perhaps it should?

Line 76: insert "lack of" before "locally"

Table 1: can it all fit on 1 page? Breaking it makes it unclear.

Methodology:

It would gain from being reworked. Sentence 1, line 128, does not add anything and could be removed. A little bit of context about what is hoping to be gleaned from what piece of data or procedure would be helpful. Does the reader really need to know precise dates and time of focus group discussions? Or would it be better to have a sense of what was discussed and how it might fit in with other pieces of data. It would help the reader to read an argument of how the methodology supports the intent. For example, sentences like "In order to examine x, we conducted a focus group discussion, which will be discussed in the findings."

Line 141: is "axe" the right word? should it be "stage" or "axis"?

NZEB Principles

Please rework lines 243-5. Also, please avoid splitting Table 2 across two pages.

A Framework for a New Standard for Chile

It is worth defining early on in the section what is meant by "high-tech" vs "low tech" NZEB. Does low-tech simply mean passive? Does high-tech mean it uses a Building Management System? Perhaps Table 3 should be at Line 267, so it is closer.

Line 273: how is "narrow temperature range" defined? This might benefit from a visual.

Lines 274-6 mention stringent comfort models for high-tech; please discuss these (is "stringent" good or bad, "stringent" relative to what). Line 279: socio-economic status, perhaps, but also human physiology. 

Lines 281-2: mentions "focus on energy performance can lead to health and comfort problems". Please elaborate: how and what, and what are the key sources? Would climate maps be useful to include?

Line 289: although they are fairly well known, it is worth defining Active House and Passive House, perhaps using a table.

Table 3: define EE and RES.

Line 305: what are "plug-loads"? Are these the same as "electrical outlets"?

Line 311: delete "the"

Line 318: delete "the"

Line 319: "emissions"

Line 320: what does "at the source" mean? Does this have to do with losses between generation and consumption? How do you measure the losses?

Line 321: should "source" be "net"?

Lines 345-57: it should be "CO2" typically.

Focus Group Discussions

Line 374: change "need to line up" to "needs to align"

Line 376: insert "There is a" before "Lack"

Line 378: delete "Limited"

Line 379: after "buildings" add "is limited"

Line 380: insert "There is an" before "underdeveloped"

Discussion

The discussion in general would benefit from more context: what and why. For example, why is there a serious need in Chile now? Is it the number of buildings being constructed over the next x years? Is it the rate of urbanisation? Is it the amount of operational energy consumed by the built environment?

Can the framework be put into a table?

Line 450-1: please articulate the current trends and future challenges.

Line 471-2: initial costs have been mentioned in other sections. Consider grouping sentences on similar topics together so the paper reads more cohesively.

Conclusions

Line 510-1: more needs to be said about the relationship to "building scale" and "environmental performance".

Author Response

-Reviewer 3

-General

The paper addresses an interesting topic, perhaps especially so given the nature of urbanisation and building. At the moment, however, the context is missing, and the paper suffers from a lack of detail.

To that end, please discuss the findings more. The paper would read better with more elaboration of key and contentious points. The narrative for why a standard is needed, how the methodology helps in this quest, and what the framework is and does vis-a-vis health and energy issues needs to be strengthened. The paper would also gain from being more direct, e.g. instead of "This paper will show that a framework for NZEB in Chile is necessary...", try "A framework for NZEB in Chile is necessary..."

Thank you very much Reviewer 3. We appreciate your time and effort and we are very happy to receive such a constructive feedback. We did our best to address all your comments. Please follow our modifications explained point by point

-English Language and Style:

Abstract needs to be thoroughly checked for capitalisation and punctuation, e.g. at the beginning of a sentence, for acronyms and proper names, and for chemical symbols (e.g. CO2). Other sections require a proofread for spelling and punctuation (e.g. double period in line 217). Could the acronyms be limited to heavily used ones in the paper? For example, FGD could simply be "focus group discussion" as it is used in only a few places. Limiting the acronyms would make for a more readable paper.

Please ensure the Bibliography follows the Journal style for italicisation, punctuation and capitalisation.

Thank you. We realize that the wording and punctuations and use of abbreviation is not consistent.  We agree with your comment and revised the whole text according to your comments. We just decided to leave the FGD abbreviation in the methodology section to avoid recurring repetition. However, outside the methodology we just avoided abbreviations to make the text more accessible for readers. See changes across the text.

-Introduction:

Table 1 should align with the discussion in the text. For example, the text discusses France, but France is not in Table 1. The text discusses auxiliary energy and heating and cooling energy, yet Table 1 does not break it down that way. Perhaps it should?

Line 76: insert "lack of" before "locally"

Table 1: can it all fit on 1 page? Breaking it makes it unclear.

We agree with the reviewer. Thank you. We omitted the USA from the text and modified Table 1 by adding France and Spain. See changes in Section 1, paragraph 1 and Table 1 and Section 3.4 and Table 1.

-Methodology:

It would gain from being reworked. Sentence 1, line 128, does not add anything and could be removed. A little bit of context about what is hoping to be gleaned from what piece of data or procedure would be helpful. Does the reader really need to know precise dates and time of focus group discussions? Or would it be better to have a sense of what was discussed and how it might fit in with other pieces of data. It would help the reader to read an argument of how the methodology supports the intent. For example, sentences like "In order to examine x, we conducted a focus group discussion, which will be discussed in the findings."

Line 141: is "axe" the right word? should it be "stage" or "axis"?

We agree with the reviewer. Thank you. We addressed the comment regarding the methodology rigor. We amended the focus group discussions description with detailed information regarding the data collection process and data quality. We indicated the background, structure, procedure and quality assurance measures of the focus group discussion. Also, we added the focus group discussion report in Appendix A. Changes can be found in Section 2.

-NZEB Principles

Please rework lines 243-5. Also, please avoid splitting Table 2 across two pages.

We agree with the reviewer. Thank you. We addressed all the comments. Changes are found in in Section 3.2 and Table 2.

-A Framework for a New Standard for Chile

It is worth defining early on in the section what is meant by "high-tech" vs "low tech" NZEB. Does low-tech simply mean passive? Does high-tech mean it uses a Building Management System? Perhaps Table 3 should be at Line 267, so it is closer.

Line 273: how is "narrow temperature range" defined? This might benefit from a visual.

Lines 274-6 mention stringent comfort models for high-tech; please discuss these (is "stringent" good or bad, "stringent" relative to what).

Line 279: socio-economic status, perhaps, but also human physiology.

Lines 281-2: mentions "focus on energy performance can lead to health and comfort problems". Please elaborate: how and what, and what are the key sources? Would climate maps be useful to include?

Line 289: although they are fairly well known, it is worth defining Active House and Passive House, perhaps using a table.

Table 3: define EE and RES.

Line 305: what are "plug-loads"? Are these the same as "electrical outlets"?

Line 311: delete "the"

Line 318: delete "the"

Line 319: "emissions"

Line 320: what does "at the source" mean? Does this have to do with losses between generation and consumption? How do you measure the losses?

Line 321: should "source" be "net"?

Lines 345-57: it should be "CO2" typically.

We agree with the reviewer. Thank you. We define the high-tech approach and moved the Table 3 earlier in the discussion. Also, we added a new Figure 3 to explain better the thermal comfort questions. Also, we explained the meaning of ‘stringent’ and re-wrote this paragraph. We kept the word ‘plug-loads’ because this is the most commonly word in the US and UK. Also, we addressed all other text improvement the comments. Changes are found in in Section 3.3 and new Figure 3.

-Focus Group Discussions

Line 374: change "need to line up" to "needs to align"

Line 376: insert "There is a" before "Lack"

Line 378: delete "Limited"

Line 379: after "buildings" add "is limited"

Line 380: insert "There is an" before "underdeveloped"

We agree with the reviewer. Thank you. We addressed all the comments. Changes are found in in Section 3.4.

-Discussion

The discussion in general would benefit from more context: what and why. For example, why is there a serious need in Chile now? Is it the number of buildings being constructed over the next x years? Is it the rate of urbanisation? Is it the amount of operational energy consumed by the built environment?

We agree with the reviewer. Thank you. We rewrote the discussion part and addressed the Chilean context more widely. Changes can be found in Section 4.1  and 4.3.

-Can the framework be put into a table?

We agree with the reviewer. Thank you. We created a new Figure and rewrote the discussion part and addressed the framework. Changes can be found in Section 3 and 4 and Figure 3.

Line 450-1: please articulate the current trends and future challenges.

Thank you. We omitted this sentence.

Line 471-2: initial costs have been mentioned in other sections. Consider grouping sentences on similar topics together so the paper reads more cohesively.

We agree with the reviewer. Thank you. We rewrote the discussion part, however we did not address the cost part.  We tried to focus on our novel framework strength and weakness and future implications. We mentioned articulated the importance of embodied energy in relation to operational energy. Also, we rearticulated the major findings and contribution of the research. Changes can be found in Section 4.

- Conclusions

Line 510-1: more needs to be said about the relationship to "building scale" and "environmental performance".

We agree with the reviewer regarding the environmental performance. Thank you. We addressed this comment in Section 5.

Reviewer 4 Report

Great article and necessary, although my final mark is major revisions.

I am not very sure about the title "Net Zero Buildings - A Framework for a New Standard for Chile", perhaps "Net Zero Buildings - A Framework for opportunities in Chile".

"The study identifies the need to advance public policies to achieve an interaction with the national electricity grid and the necessity to create a calculation and weighting system that allows counting the primary energy within the energy neutrality balance concept". There are many articles about this. It is no necessary in this article do it again.

L.41. United States as reference for NZEB?

L.48. France is not in Table 1.

Table 1. Explain the order of countries.

L.69-76. Spain is a reference in saving energy in buildings. This must be take into account.

"As such, the framework brings a consensus for best practices in European countries regarding NZEB, to bridge the knowledge gap and to eventually increase the NZEB market uptake in Chile and potentially in Latin America". In my opinion, this is the key point of the project which is not correctly developed (I insist, in my opinion). Please, it is not necessary to create new methodologies, 'only' to translate existing ones to the possibilities for Chile. Experiences such as Prof-Trac or equivalents MUST be a reference in your article and future work. More details at http://proftrac.eu/open-training-platform-for-nzeb-professionals.html

L.121. 40 publications? There are only 31 references at the end. This search of related publications must be improved. In fact, I am sure that doing it, authors will find more related works.

L.157. FGD was working 'only' 1 day from 2 to 7 AM?

L.209-213. Very well.

L.293. Ceiling fans - OK. Mobile stoves - Never? Discuss about this please.

Table 3. 1200 ppm. Why? I don't agree.

L.385-387. These lines to discussion?

L.507. Develop a formula + some considerations, although helpful, in my opinion is not a framework. This is why I argue to, perhaps, change the title.

Chapter 4. It has not clear structure for me.

Chapter 5. Concrete solutions, do not repeat the possibilities.

Author Response

-Reviewer 4

Great article and necessary, although my final mark is major revisions.

Thank you very much Reviewer 4. We appreciate your time and effort and we are very happy to receive such a constructive feedback. We did our best to address all your comments. Please follow our modifications explained point by point.

I am not very sure about the title "Net Zero Buildings - A Framework for a New Standard for Chile", perhaps "Net Zero Buildings - A Framework for opportunities in Chile".

We agree with the reviewer. Thank you. We changed the paper title.  

"The study identifies the need to advance public policies to achieve an interaction with the national electricity grid and the necessity to create a calculation and weighting system that allows counting the primary energy within the energy neutrality balance concept". There are many articles about this. It is no necessary in this article do it again.

We agree with the reviewer. Thank you. We shortened the text. See changes in Section 1.  

L.41. United States as reference for NZEB?

L.48. France is not in Table 1.

Table 1. Explain the order of countries.

L.69-76. Spain is a reference in saving energy in buildings. This must be take into account.

We agree with the reviewer. Thank you. We omitted the USA from the text and modified Table 1 by adding France and Spain.. See changes in Section 1, paragraph 1 and Table 1.  

"As such, the framework brings a consensus for best practices in European countries regarding NZEB, to bridge the knowledge gap and to eventually increase the NZEB market uptake in Chile and potentially in Latin America". In my opinion, this is the key point of the project which is not correctly developed (I insist, in my opinion). Please, it is not necessary to create new methodologies, 'only' to translate existing ones to the possibilities for Chile. Experiences such as Prof-Trac or equivalents MUST be a reference in your article and future work. More details at http://proftrac.eu/open-trainingplatform-for-nzeb-professionals.html

We agree with the reviewer. Thank you. We rearticulated this text and added the Prof-Trac reference and amended it with two other recent and similar references. See changes in Page 4, paragraph 1.  

L.121. 40 publications? There are only 31 references at the end. This search of related publications must be improved. In fact, I am sure that doing it, authors will find more related works.

We agree with the reviewer. Thank you. We amended the reference with 10 new references to reach 40.  

L.157. FGD was working 'only' 1 day from 2 to 7 AM?

L.209-213. Very well.

L.293. Ceiling fans - OK. Mobile stoves - Never? Discuss about this please.

Table 3. 1200 ppm. Why? I don't agree.

We agree with the reviewer. Thank you. We addressed all comments above in the text. We changed the 1200 ppm in Table 3 because this was the recommendation of the focus group experts.  

L.385-387. These lines to discussion?

We agree with the reviewer. Done.

L.507. Develop a formula + some considerations, although helpful, in my opinion is not a framework. This is why I argue to, perhaps, change the title.

We agree with the reviewer. Thank you. We changed the paper title. Also we added a new Figure illustrating the framework and provided a description in Section.  

Chapter 4. It has not clear structure for me.

We agree with the reviewer. Done.

Chapter 5. Concrete solutions, do not repeat the possibilities.

We agree with the reviewer. Done.

Round 2

Reviewer 1 Report

The new version of the article has adequately incorporated all the comments made by this reviewer.

Reviewer 2 Report

The authors have made some major improvements to the paper and strenghtened is methodological and theorical rigor. I am happy to recommend full acceptance of this revised version.

Reviewer 4 Report

Comments from reviewer have been included.